# Beyond Correctness: Harmonizing Process and Outcome Rewards through RL Training

## Abstract

Reinforcement learning with verifiable rewards (RLVR) has emerged as a predominant paradigm for mathematical reasoning tasks, offering stable improvements in reasoning ability. However, Outcome Reward Models (ORMs) in RLVR are too coarse-grained to distinguish flawed reasoning within correct answers or valid reasoning within incorrect answers. This lack of granularity introduces noisy and misleading gradients significantly and hinders further progress in reasoning process quality. While Process Reward Models (PRMs) offer fine-grained guidance for intermediate steps, they frequently suffer from inaccuracies and are susceptible to reward hacking.

To resolve this dilemma, we introduce PRocess cOnsistency Filter (PROF), an effective data process curation method that harmonizes noisy, fine-grained process rewards with accurate, coarse-grained outcome rewards. Rather than naively blending PRM and ORM in the objective function (Zou et al., 2025), PROF leverages their complementary strengths through consistency-driven sample selection. Our approach retains correct responses with higher averaged process values and incorrect responses with lower averaged process values, while maintaining positive/negative training sample balance. Extensive experiments demonstrate that our method not only consistently improves the final accuracy over $4\%$ compared to the blending approaches, but also strengthens the quality of intermediate reasoning steps.

## 1 Introduction

Verifiable rewards have spurred the widest attention recently because they can reliably improve the performance on reasoning tasks with easily verifiable outcomes, such as mathematical and coding problems (Cobbe et al., 2021; Jaech et al., 2024; Shao et al., 2024; Xiong et al., 2025b). However, since the verifiers can only verify the outcome results, the rewards are too sparse and coarse to measure and supervise the reasoning quality in intermediate steps. For instance, if a correct answer contains flawed logic, Outcome Reward Models (ORMs) cannot distinguish it from a completely correct response. We present a classic example from the training data in Table 1, which has fundamentally invalid reasoning but happens to obtain the correct answer. Incorporating such flawed examples into the training process introduces unreliable gradients, leading to significant instability and misguided learning. Moreover, the quality and interpretability of the Chain of Thought (CoT) are crucial for the practical reasoning ability of a model, not just the accuracy of the final answer (Zhu et al., 2025; Lyu et al., 2023; Yeo et al., 2024). The lack of faithfulness during CoT is also observed by (Baker et al., 2025; Chen et al., 2025b), limiting its applications in areas such as LLM safety monitoring and interpretation.

Hence, the limitation of ORMs can be partially addressed by using LLM-as-a-judge or Monte-Carlo (MC) estimation to provide step-wise judgments or values (Wang et al., 2023; Zheng et al., 2024). However, the cost of inferring LLM step-wise judgments or MC estimation at each iteration during online training

---

**Problem:** There are $4$ coins of $1, 2, 3, 5$ cruzeiros, which weigh $1, 2, 3, 5$ grams respectively. One of them is counterfeit, differing in weight (but not known whether it is heavier or lighter). How can the counterfeit coin be identified with the minimum number of weighings using a balance scale without weights?

---

**Step 1: Introduction.** We need to identify a counterfeit coin with an unknown weight difference from ...

**Step 2 - 6: The Flawed Weighing Logic and Code.**
1. Weigh coins of **1 gram and 2 grams** against coins of **3 grams and 5 grams**. 2. Based on the result of the first weighing: - If the left side is heavier, then the counterfeit coin is either 1 gram or 3 grams. - If the right side is heavier, then the counterfeit coin is either 2 gram or 5 grams. - If both sides are equal, then the counterfeit coin is either 2 gram or 5 grams. 3. For the second weighing, ...
Let's illustrate this with code and ensure that the steps are correct ... output: (1, 'heavier').
**Step 7, 8, 9: Summary and Final Answer.**
Here is a summary of the steps: 1. Weigh coins of 1 gram and 2 grams against coins of 3 grams and 5 grams...
2. For the second weighing, weigh the 1 gram coin against a known genuine coin (3 grams)...
Thus, the minimum number of weighings required to identify the counterfeit coin is $\boxed{2}$ .

Table 1: An Example of Reaching a Correct Result Through Flawed Reasoning. The proposed first weighing of {1g, 2g} (total 3g) against {3g, 5g} (total 8g) is fundamentally **invalid**. A balance scale requires comparing groups of equal nominal weight. Because this weighing is unbalanced, all conclusions drawn from it are baseless. The final answer is correct but is completely unsupported by the fallacious reasoning.

is so high. Hence, it is inefficient and expensive to infer the step-wise scores or values for online training. Alternatively, an efficient solution is to use the pre-trained Process Reward Models (PRMs) (Lightman et al., 2023; Zhang et al., 2025). However, applying these models to the online training process often suffers from misspecification and distribution shift due to the limitations of offline training data. Especially in boundary cases where the policy encounters difficult problems and produces rarely seen responses, PRMs often fail to judge them correctly, thus leading to severe reward hacking (Michaud et al., 2020; Tien et al., 2022). Even if some works (Zha et al., 2025; Cui et al., 2025) attempt to co-train the policy and PRMs online, they can only train in implicit ways such as using implicit generative reward or aligning process rewards with outcomes.

Although numerous works have made enormous efforts to train PRMs offline or online, the problem of effectively coordinating PRMs with outcome-verifiable rewards remains largely underexplored. Existing approaches typically combine process and outcome rewards in a simple weighted manner (Zha et al., 2025; Cui et al., 2025; Zou et al., 2025), which is vulnerable for reward hacking due to the noises and misspecification in PRMs. Therefore, in this paper, instead of developing another PRM, we focus on how to robustly integrate a pre-trained PRM into the online training process, i.e.,

*H*ow to harmonize the accurate but coarse-grained ORMs with fine-grained but noisy Process Reward
Models (PRMs) in Reinforcement Learning (RL)?

In this work, instead of fine-tuning another PRM, we answer this question with a **PRocess cOnsistency Filtering (PROF)** framework, a data curation strategy based on process-outcome consistency. PROF over-samples more responses at training time, and then, ranks and filters the responses by the consistency between their PRMs and ORMs. Specifically, it removes samples where the process and outcome signals conflict—such as correct responses derived from flawed reasoning, or incorrect responses that contain sound reasoning steps. By filtering out these inconsistent samples, PROF eliminates conflicting and noisy gradients. Furthermore, observing that correct and incorrect responses have different consistency distributions, we rank each group separately to maintain a balanced training ratio. PROF is a modular framework that can be combined with RL algorithms like Group Relative Policy Optimization (GRPO) for online training.

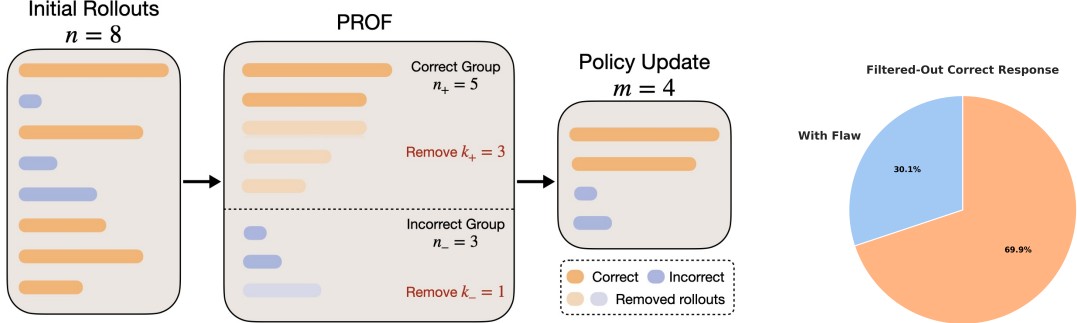

Figure 1: Left: Visualization of PROF Algorithm 1, where the length of each rectangle represents values of process rewards averaged over steps for each rollout. After generating $n$ rollouts and process rewards, PROF ranks the correct and incorrect group separately according to PRM-ORM consistency, so for the correct group, the longer items are kept; for the incorrect group, the shorter items are kept. The number to remove is to balance correct and incorrect ratio. Right: Fraction of flawed-reasoning responses judged by LLM among the filtered-out correct responses.

We conduct extensive experiments to validate the improvement of PROF-GRPO on both outcome accuracy and process reasoning quality at diverse math reasoning benchmarks using both Qwen (Yang et al., 2024) and LLaMA (Dubey et al., 2024) models. To summarize, we highlight our key contributions as follows:

- We propose **PRocess cOnsistency Filtering (PROF)** to robustly integrate noisy Process Reward Models (PRMs) with Outcome Reward Models (ORMs). Compared to the GRPO-type algorithms that only leverage outcome rewards, our implementation PROF-GRPO can effectively distinguish the inconsistent trajectories, such as correct answers with flawed reasoning steps or incorrect answers with mostly valid steps. Moreover, unlike prior approaches that simply blend PRMs and PRMs, our method only rely on PRMs to rank and filter rather than directly involving them into gradients. This separation essentially avoids reward hacking and entropy collapse, thus achieving stable performance gains throughout training.

- We conduct extensive studies to demonstrate that PROF-GRPO not only increases the final outcome accuracy but also shapes the intermediate reasoning steps and improves the process reasoning quality. Various metrics such as Monte-Carlo estimation, LLM-as-a judge are used to validate that our method enable models to segment reasoning trajectories into **detailed and easy-to-verify** steps.

- We conduct a series of ablation studies to illustrate the importance of separating the correct and incorrect responses during the filtration. Meanwhile, we investigate various ways of calculating the consistency and filtering, and ablate on LLaMA base models for generalization.

## 2 RELATED WORK

**Sample Filtering in Reinforcement Learning for LLM.** A key challenge in applying reinforcement learning to LLM applications is the imperfection of reward signals. These signals stem from a learned reward model, such as Reinforcement Learning from Human Feedback (RLHF), or are sparse, delivered only at the end of a trajectory (e.g. RLVR). In RLHF, the reward model is trained on human-annotated pairwise comparisons, typically using a Bradley-Terry model (Bradley & Terry, 1952). Due to inherent human disagreement and finite training data, the model develops shortcuts that RL algorithms can exploit (Lin et al., 2023; Eisenstein et al., 2023) to chase for a fake high reward. Consequently, these rewards may not fully align with the underlying intended goals, leading to reward hacking.

Data filtering, a data curation technique, has proven effective in mitigating this issue across various LLM applications with RL. A prominent line of work proposes filtering training pairs based on the reward gap between the chosen and rejected responses (Yuan et al., 2024; Dong et al., 2024; Xiong et al., 2024a; Zhang et al., 2024). The high-level intuition is that a larger reward gap indicates higher model confidence, making these pairs less noisy and more reliable for training when the reward model is well-calibrated. Moreover, Kim et al. (2024); Yu et al. (2025a) further rank and filter the samples by combining their rewards and responses length during the preference learning process.

In RLVR, where rewards are sparse and only for the outcome, filtering is also helpful. For instance, the simple rejection sampling fine-tuning (Dong et al., 2023; Chen et al., 2025a), which discards all incorrect trajectories, often approaches the performance of more complex algorithms like GRPO (Dong et al., 2023; Chen et al., 2025a; Xiong et al., 2025a). Other methods like (Yang et al., 2024) filter prompts by difficulty prior to the RL training. Yu et al. (2025b) removes prompts that yield zero gradients during training and dynamically regenerates samples. This technique is known as dynamic sampling and has been rather widely accepted. Xiong et al. (2025a) demonstrates that prompts where all generated responses are incorrect can significantly hurt the performance of the vanilla Reinforce algorithm. They propose an online data filtering strategy based on the correctness reward, showing that a modified Reinforce with filtering (Reinforce-rej) can match or exceed GRPO's performance. Their results suggest that the advantage of GRPO compared to Reinforce is due to the implicit data filtering mechanism from the reward shaping. Finally, Xu et al. (2025) proposes to over-sample and keep a subset such that the variance of the rewards in the subset is maximized, which implies that they try to balance the ratio of correct and incorrect responses for reasoning tasks.

In contrast to these methods, which primarily rely on coarse, outcome-based metrics (e.g., final answer correctness, trajectory-level rewards), our approach introduces a more fine-grained filtering mechanism. We leverage process-supervised reward models (PRMs) (Lightman et al., 2023) to evaluate and filter based on the quality of intermediate reasoning steps, and their consistency with ORMs.

## 3 METHOD

An LLM is a policy distribution such that given a prompt $x$, it provides the density $\pi(a|x)$ of generating each response $a$. For mathematical reasoning tasks with a binary verifiable reward, there exists a verifier mapping prompt-response pairs $(x, a)$ to a scalar reward $r_o(x, a) \in \{-1, 1\}$. For each prompt, we can generate a group of responses and their corresponding responses with the verifier $\{(a_i, r_{o,i})\}_{i=1}^{G}$.

**GRPO.** (Shao et al., 2024) proposes this policy gradient algorithm that simplifies the Proximal Policy Optimization (PPO) (Schulman et al., 2017) by only computing the advantage based on the outcome rewards in a group. Instead of maintaining and updating another value network, GRPO computes the advantage by standardizing the outcome rewards within a group:

$$A_i = \frac{r(x, a_i) - \text{mean}\left(\{r(x, a_j)\}_{j=1}^{n}\right)}{\text{std}\left(\{r(x, a_j)\}_{j=1}^{n}\right) + \delta}, \quad i = 1, \ldots, n,$$

where $r(x, a_i)$ is the reward for a given response and $\delta > 0$ is a small constant for numerical stability. Let $a_t$ denote the $t$-th token of response $a$ and $a_{<t}$ denotes $(a_1, \ldots, a_{t-1})$. This advantage is then incorporated into a clipped surrogate objective function, which is optimized to update the policy from $\pi_{\theta_{\text{old}}}$ to $\pi_{\theta}$:

$$\mathcal{J}_{\text{GRPO}}(\theta) = \mathbb{E}_{x \sim \mathcal{D}} \left[ \frac{1}{n} \sum_{i=1}^{n} \frac{1}{|a_i|} \sum_{t=1}^{|a_i|} \min \left( \frac{\pi_{\theta}(a_{i,<t}|x)}{\pi_{\theta_{\text{old}}}(a_{i,<t}|x)} A_i, \text{clip}\left( \frac{\pi_{\theta}(a_{i,<t}|x)}{\pi_{\theta_{\text{old}}}(a_{i,<t}|x)}, 1 - \epsilon, 1 + \epsilon \right) A_i \right) \right].$$

Although this approach stabilizes the online policy optimization and is efficient, the sparse reward signal limits further improvement on the intermediate reasoning steps.

---

**Algorithm 1** Process Consistency Filter (PROF)

---

1: **Input:** Number of rollouts $n$, policy update size $m$, rollout $\{a_1, \ldots, a_n\}$, outcome rewards $\{r_{o,1}, \ldots, r_{o,n}\}$, step number regularization parameter $\lambda$, $H_\lambda > 0$.

2: Obtain process rewards for each rollout $a_i$ with $H_i$ steps: $(r_i^1, \ldots, r_i^{H_i})$ and compute trajectory-wise consistency

$$r_i^{\text{pro}} = \Big[ \frac{1}{H_i} \sum_{h=1}^{H_i} r_i^h - \lambda I(H_i = 1 \text{ or } H_i \geq H_\lambda) \Big] \cdot r_{o,i}. \tag{1}$$

3: Divide rollouts into correct group $\mathcal{G}_+ = \{a_1^+, \ldots, a_{n_+}^+\}$ with $r_{o,i} = 1$ and incorrect group $\mathcal{G}_- = \{a_1^-, \ldots, a_{n_-}^-\}$ with $r_{o,i} = -1$, where $n_+ + n_- = n$.

4: Compute kept number $k_+ \in [n_+], k_- \in [n_-]$ in each group such that $K_+ + k_- = m$ and $k_+ k_-$ is maximized.

5: Rank $\mathcal{G}_+$ and $\mathcal{G}_-$ by $r^{\text{pro}}$ separately, and keep the samples

$$\mathcal{K}^+ = \{a_i^+ | \text{rank}(a_i^+) \geq n_+ - k_+\}, \quad \mathcal{K}^- = \{a_i^- | \text{rank}(a_i^-) \geq n_- - k_-\}.$$

6: **Output:** The kept trajectories $\mathcal{K}^+ \cup \mathcal{K}^-$ with final kept size $m$.

---

**Process Reward Model (PRM).** For a response $a$ composed of multiple reasoning steps $a = (a^1, \ldots, a^H)$, we follow previous works (Zheng et al., 2024; Zhang et al., 2025; Zou et al., 2025) to use a newline as a sign for a new step. For each step $a^h$, the PRM $r^h$ maps it, the previous steps and the prompt $(x, a^{\leq h})$ to a scalar $r^h(x, a^{\leq h})$, where we use the short-hand notation $a^{\leq h} = (a^1, \ldots, a^h)$.

**Our Method PROF: Process Consistency Filter Framework**   We propose PROF in Algorithm 1 to incorporate the consistency of PRMs and ORMs robustly after the rollout phase, and also present a visualization in Figure 1. First, we generate $G$ samples and get the outcome reward. Then, we call the PRM to generate step-wise rewards for each rollout and compute the trajectory-wise consistency score $r^{\text{pro}}$ by taking the mean over the step-wise rewards and adding a step length regularization in equation 1, where $\lambda$ is the regularization parameter and $H_\lambda$ is the threshold for the penalized step number. This regularization is to ensure that samples with no step segments or over-long steps are discarded in the correct group. The samples are divided into two subgroups: $\mathcal{G}_+$ contains the correct samples with $r_o = 1$, and $\mathcal{G}_-$ contains the incorrect samples with $r_o = -1$. Inspired by (Xu et al., 2025), the numbers to discard in each subgroup $k_+, k_-$ are calculated to maximize the outcome-reward variance of the final kept samples $k_+ k_- / (k_+ + k_-)^2$. Since $k_+ + k_- = m$ is fixed, $k_+ k_- = k_+(m - k_+)$ should be maximized and the maximum is obtained when $k_+$ is closest to $m/2$ under the constraint $k_+ \leq n_+, k_- \leq n_-$. This implies that the ratio of correct and incorrect responses should be balanced. After that, we use $r^{\text{pro}}$ to rank and filter the correct group and randomly filter the incorrect group. Finally, we collect the kept $m$ trajectories for policy update.

## 4 EXPERIMENTS

### 4.1 SETUP

We focus on mathematical reasoning tasks in this work. For online training, we use the prompt set Numina-Math (Beeching et al., 2024) containing nearly 860k math problems with ground-truth answers ranging from Chinese high school math exercises to US and international mathematics Olympiad competition problems. We choose Qwen2.5-Math-1.5B-base, Qwen2.5-Math-7B-base (Yang et al., 2024) as the training base models. For the PRM, we use Qwen2.5-Math-PRM-7B (Zhang et al., 2025) to generate process rewards. More details are provided in Appendix C. The models' performance is evaluated on 5 benchmarks: Math500 (Hendrycks et al., 2021), Minerva Math (Lewkowycz et al., 2022), Olympiad Bench (He et al., 2024),

AMC2023[1] and AIME2024[2]. We mainly use average@16 for evaluation, i.e., the accuracy is averaged over 16 responses per prompt under temperature 1.0. The models are allowed to generate 4096 tokens.

## 4.2 MAIN RESULTS

| Model | Algorithm | Math500 | Minerva Math | Olympiad Bench | AIME24 | AMC23 | Average |
|---|---|---|---|---|---|---|---|
| Qwen2.5-Math-1.5B-base | Base | 39.9 | 11.4 | 19.1 | 3.5 | 23.6 | 19.5 |
| | GRPO | 70.3 | 29.1 | 33.0 | 9.0 | 44.5 | 37.2 |
| | Blend | 67.6 | 27.8 | 31.1 | 7.7 | 42.5 | 35.3 |
| | PROF-GRPO | 73.2 | 30.0 | 36.1 | 9.6 | 49.1 | 39.6 |
| Qwen2.5-Math-7B-base | Base | 42.0 | 12.8 | 19.2 | 12.9 | 30.0 | 23.4 |
| | GRPO | 81.6 | 37.2 | 45.5 | 20.6 | 64.4 | 49.9 |
| | Blend | 81.7 | 36.7 | 45.0 | 15.2 | 58.0 | 47.3 |
| | PROF-GRPO | 83.1 | 39.0 | 47.8 | 17.5 | 70.9 | 51.7 |

Table 2: Performance of different algorithms across five benchmarks including Math500 (Hendrycks et al., 2021), Minerva Math (Lewkowycz et al., 2022), Olympiad Bench (He et al., 2024), AMC2023 and AIME2024. We denote Blend-PRM-GRPO by Blend for short. We tune all the algorithms to their best performance. The reported accuracy is average@16 under temperature 1.0.

We summarize our main results in Table 2, where Blend denotes a common way that mixes the PRM with outcome rewards (Zha et al., 2025; Cui et al., 2025; Zou et al., 2025). Following (Zou et al., 2025), the PRMs are averaged over steps for each response, weighted by a parameter $\beta$, and added to outcome rewards. We use parameter $\beta = 0.8$ according to Table 5 of (Zou et al., 2025). Our main findings are as follows.

**PROF-GRPO Outperforms the Baselines.** As shown in Table 2, our proposed method, PROF-GRPO, consistently outperforms GRPO and Blend-PRM-GRPO over various benchmarks. Specifically, for models starting from Qwen2.5-Math-1.5B-base, PROF-GRPO achieves an average accuracy of 39.6%, surpassing the standard GRPO baseline (37.2%) and the Blend-PRM-GRPO method (35.3%). A similar trend is observed with the Qwen2.5-Math-7B-base model, where PROF-GRPO achieves a 51.7% average accuracy, a significant improvement over GRPO's 49.9% and Blend-PRM-GRPO's 47.3%. [3] The learning dynamics in Figure 2 corroborate these findings, illustrating that PROF-GRPO steadily maintains a consistent performance advantage over both GRPO and Blend-PRM-GRPO throughout the training process. Notably, PROF-GRPO achieves faster convergence rate and higher final accuracy than GRPO.

**Filtration Method is Much More Robust than Blending.** We plot the entropy loss and response length curves of GRPO, Blend-PRM-GRPO and PROF-GRPO in Figure 2. Blending-PRM-GRPO suffers from severe rewarding hacking since its entropy collapses quickly towards zero. Simultaneously, its response length in the right figure uncontrollably increases, indicating that the model has learned to game the PRM by over-generating verbose responses and more repetitive steps to get a higher averaged process reward. Therefore, Blend-PRM-GRPO's testing accuracy even falls below GRPO. In contrast, PROF-GRPO maintains gradual and slightly faster decrease in entropy loss and controllable response length growth. This illustrates that our filtration method effectively leverages the PRM signal while stay robust to reward hacking. We will carefully analyze and compare the quality of intermediate reasoning steps of our method and baselines.

## 4.3 HOW PROF SHAPES INTERMEDIATE REASONING STEPS

**Effectiveness of Consistency Filtration.** To demonstrate that our algorithm effectively differentiates the inconsistent trajectories, especially those correct answers with flawed reasoning steps, we prompt Qwen2.5-Math-7B-base (Yang et al., 2024) to generate rollouts for 500 problems randomly selected from the training

---

[1] https://huggingface.co/datasets/math-ai/amc23

[2] https://huggingface.co/datasets/math-ai/aime24

[3] Although PROF-GRPO underperformed GRPO on AIME24 for Qwen2.5-Math-7B-base, given the dataset's small size of only 30 samples, the performance difference may not be statistically significant.

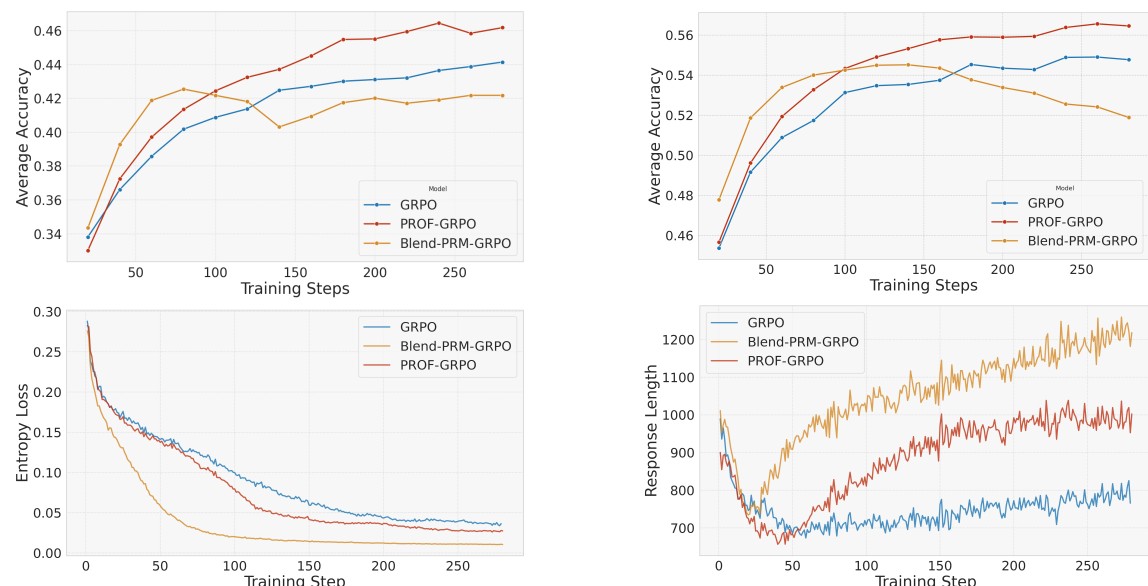

Figure 2: The learning dynamics of PROF-GRPO initialized from Qwen2.5-Math-1.5B-base (upper left) and Qwen2.5-Math-7B-base (upper right) in comparison of GRPO and Blend-PRM-GRPO. The y-axis is the average@16 accuracy and is further averaged on Math500, Minerva Math and Olympiad Bench. Entropy loss (lower left) and response length (lower right) of the models initialized from Qwen2.5-Math-7B-base.

set, and implement the filtration in Algorithm 1. Then, the filtered-out correct responses are judged by Claude-3-7-sonnet from Anthropic to verify whether they contain flawed steps. We use the prompt in Zhang et al. (2025) and provide the details in Appendix C. From Figure 1, 30.1% responses among the filtered-out correct responses are judged to possess flawed reasoning. This indicates that our methods can efficiently distinguish a number of flawed responses and reach consensus with LLM. Furthermore, with human checking those filtered-out correct responses, there are many responses with invalid or even completely wrong reasoning steps but luckily reaching the correct answer. A typical example is presented in Table 1. However, such flawed reasoning processes would be entirely missed by a standard ORM.

**Improved Step-wise Value.** To evaluate the quality of intermediate steps, we adopt Monte Carlo (MC) estimation, a common way to estimate probability of getting to correct final answers (Wang et al., 2023; Xiong et al., 2024a; Luo et al., 2024). For this analysis, we select problem-response pairs from the test prompts where our method (PROF-GRPO) and GRPO both produced the correct final answer. Both models were initialized from Qwen2.5-Math-7B-base. To estimate the value of each reasoning step, we generate eight independent completions from that point using a temperature of 1.0, and the resulting empirical success rate serves as the MC value. Our primary finding is that PROF-GRPO achieves significant improvement in step-wise values compared to GRPO. In Figure 3, the average MC estimations across all five benchmarks are consistently higher for our model. The specific improvement gaps are 9.2% on Math500, 37.4% on Minerval Math, 15.9% on Olympiad Bench, 9.2% on AMC2023, and 11.1% on AIME2024, which are much larger than the outcome accuracy gap in Table 2. Hence, in addition to improving the outcome accuracy, our PROF method substantially improves the quality and consistency of intermediate steps.

**Deeper Analysis on Math500.** We further compare responses where both models were correct on Math500 in Figure 3. In the second left figure, PROF-GRPO exhibits more reasoning steps. In the third left figure, the PRM used for training assigns higher rewards for PROF-GRPO's responses. In the right-

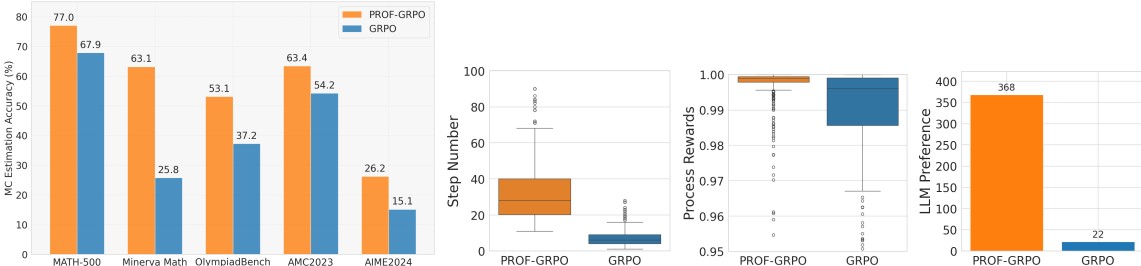

Figure 3: Reasoning intermediate-steps performance of PROF-GRPO in comparison with GRPO. The most left plot is the Monte Carlo (MC) estimation scores across five benchmarks. The other three are on Math500 under metrics of number of steps (2nd left), the averaged process rewards generated by Qwen2.5-Math-PRM-7B (3rd left), and LLM's preference between two modes' responses (most right).

most figure, we use Claude to judge which one's reasoning process has more complete and detailed steps, and PROF-GRPO's responses are significantly preferred. The prompt for LLM-as-a-judge is presented in Table C.2. The key takeaway is that our PROF method reshapes the model's CoT process from unfaithful reasoning into **detailed and easy-to-verify** steps. This is further validated by two examples in Figure 7, 8.

## 5 ABLATIONS

### 5.1 SEPARATION OF CORRECT AND INCORRECT GROUP

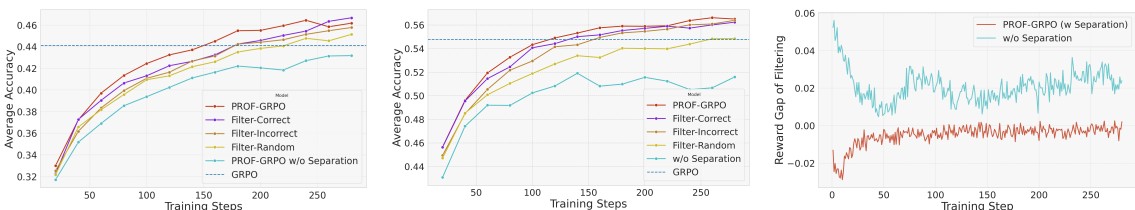

Figure 4: Left two: averaged accuracy over Math500, Minerva Math and Olympiad Bench for PROF-GRPO and its variants initialized from Qwen2.5-Math-1.5B-base and Qwen2.5-Math-7B-base. Most right: the gap between the training rewards after and before the filtering for PROF-GRPO in comparison with not separating correct and incorrect groups (w/o separation).

We conduct an ablation experiment on the necessity of separating correct and incorrect samples, named as PROF-GRPO w/o separation, where the rollouts are ranked and filtered together. To mitigate bias in PRM, each step's PRM is subtracted by the averaged PRM of the batch. Even after centering, the rightmost plot in Figure 4 shows that PROF-GRPO w/o Separation has over $2\%$ gap between the training reward after and before the filtration. This indicates that a disproportionate number of negative samples are removed. One explanation is that incorrect responses often contain several correct intermediate steps, thus increasing the averaged PRM over steps and leading to lower consistency. Consequently, incorrect responses exhibit lower consistency than correct ones, especially as the policy model improves over training. In contrast, PROF-GRPO successfully balances the bias by separating the correct and incorrect groups.

To further disentangle the contributions of filtering correct versus incorrect samples, we design the following variants of PROF: (1) Filter-Correct: use PRM consistency to filter the correct group and randomly filter the incorrect group; (2) Filter-Incorrect: only use PRM consistency to filter the incorrect group; (3) Filter-Random: randomly filter both correct and incorrect samples Xu et al. (2025). In Figure 4, Filter-Correct and

PROF-GRPO (Filter-both) achieve comparably best performances among the variants across the 1.5B and 7B models. While Filter-both converges more efficiently because it leverages the consistency filtration for both correct and incorrect groups. Filter-incorrect is less efficient and has slightly poorer performance. In contrast, Filter-Random only performs slightly better than GRPO, and w/o Separation performs the worst.

We find that separating the correct and incorrect groups is essential to prevent the over-removal of valuable incorrect samples during training. While both Filter-both and Filter-Correct are top-performing strategies, with the former being more efficient, the trade-offs between them will be discussed in the following section. Furthermore, the comparable performance of Filter-both and Filter-Correct indicates that the process quality for correct samples is more crucial than the consistency for incorrect samples during the training process.

## 5.2 Ablation Study on Base Model

| Algorithm | Math500 | Minerva Math | Olympiad Bench | AIME24 | AMC23 | Average |
|---|---|---|---|---|---|---|
| Base | 30.0 | 8.8 | 6.1 | 2.3 | 10.6 | 11.6 |
| GRPO | 50.5 | 18.8 | 17.9 | 5.0 | 25.6 | 23.6 |
| Blend-PRM-GRPO | 37.2 | 13.1 | 9.9 | 1.0 | 17.2 | 15.7 |
| PROF-GRPO (Both) | 50.4 | 19.1 | 18.7 | 3.5 | 27.8 | 23.9 |
| PROF-GRPO (Correct) | **52.4** | **19.5** | **19.8** | **6.7** | **28.6** | **25.4** |
| PROF-GRPO (Incorrect) | 49.0 | 18.0 | 17.3 | 5.4 | 23.9 | 22.7 |

Table 3: The test accuracy of different methods initialized from LLaMA-3.2-3B-instruct that is average@16 under temperature 1.0 and further averaged across all the five benchmarks.

To showcase the generalization of our algorithm, we conduct experiments on LLaMA-3.2-3B-instruct (Dubey et al., 2024) that has weaker math-reasoning abilities and more distribution shift since Qwen2.5-Math-PRM-7B is trained on the distribution of Qwen's family. As provided in Table 3, PROF-GRPO with PRM consistency filtering both correct and incorrect groups (Both) achieves 23.9%, marginally outperforming the GRPO baseline (23.6%), while only applying PRM consistency to filter the correct group (Correct) exhibits the strongest (25.4%) performance. Conversely, applying the filter solely to the incorrect group (PROF-GRPO (Incorrect)) is counterproductive, causing accuracy to drop to 22.7%. Blend-PRM-GRPO still scores the worst (15.7%) among all the methods. These results suggest that our PROF methods can consistently outperform baselines across various base models.

For the trade-off between the Both and Correct, we conclude that when the PRM is less reliable or prone to reward hacking (as in this cross-model scenario), the "Correct" method offers more robust improvements by safely constraining the PRM's influence. However, when the PRM is highly reliable and training efficiency is a priority, the "Both" method is recommended. Due to the space limit, more ablations such as rollout numbers and various filtration methods are provided in Appendix D.

## 6 Conclusion and Future Work

This work introduces Process Consistency Filter (PROF), a novel data curation technique that filters generated responses by the data PRM-ORM consistency, and maintains the balance of correct-incorrect ratios. We demonstrate its effectiveness in both consistently improving the accuracy of obtaining correct final answers and shaping the policy model to generate more detailed and fine-grained segmented intermediate reasoning steps. Particularly, PROF is a general filtration framework without reliance on specific PRMs or the RL algorithms. Thus, the use of Qwen2.5-Math-PRM-7B as the PRM in our experiments is not a limitation. Exploring the integration of PROF with more accurate or diverse PRMs remains an interesting direction for future work. Additionally, how to extend our method to other reasoning tasks, such as coding (Jimenez et al., 2023) and web navigation (Zhou et al., 2023) deserves to be explored.

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

## A  LLM Usage

We use LLM to find grammar mistakes and polish the writing.

## B  Additional Related Works

**Process-Supervised Reward Models for Fine-Grained Feedback.**  The RLHF focuses on the *trajectory-level comparison* under the Bradley-Terry model. For reasoning-related task, Yang et al. (2024) uses the correctness of the final answer to construct the preference pairs and trains Bradley-Terry reward models for mathematical reasoning. A more widely used approach, termed Outcome Reward Models (ORMs) trains a classifier to predict whether the final answer is correct or not based on the reasoning history. However, Lightman et al. (2023) have shown that Process-Supervised Reward Models (PRMs), which evaluate each intermediate step of a reasoning chain, significantly outperform ORMs, especially for data selection tasks like best-of-n sampling (Lightman et al., 2023). But their approach requires human annotators to label each intermediate steps of the reasoning. Wang et al. (2023) proposes to use Monte-Carlo estimation of the Q value to automatically decide the label. After this, a long line of works proposes to improve the PRMs by generative reward modeling, advanced training technique like RL, and refined engineering practices (Xiong et al., 2024b; Zhang et al., 2025; Khalifa et al., 2025; Zhao et al., 2025; Xiong et al., 2025c). Our work does not focus on improving PRMs but uses the PRMs to supervise the intermediate steps of CoT trajectories for data filtering. We mainly use the Qwen2.5-Math-PRM-7B from Zhang et al. (2025) as it is trained on the distribution of Qwen model and achieves superior performance on ProcessBench (Zheng et al., 2024).

## C  Additional Experimental Details and Results

### C.1  Main Experiments

The implementations are based on the verl framework (Sheng et al., 2025), and we follow most of the parameter settings in verl. Detailedly, we apply the AdamW optimizer with learning rate $1 \times 10^{-6}$. We adopt the clip higher trick (Yu et al., 2025b) that clips the sampling ratio $\pi_\theta/\pi_{\text{old}}$ to an asymmetric range $(1 - \epsilon_{\text{low}}, 1 + \epsilon_{\text{high}})$. Specifically, we set $\epsilon_{\text{low}} = 0.2, \epsilon_{\text{high}} = 0.28$ for models started from Qwen2.5-Math-1.5B-base and maintain $\epsilon_{\text{high}} = \epsilon_{\text{low}} = 0.2$ for other cases. In each iteration, we sample 1024 prompts, rollout $n = 4$ responses per prompt for GRPO and $n = 8$ responses for PROF-GRPO. Note that the policy update number for all algorithms is $m = 4$. For the regularization of step numbers in Algorithm 1, we take $\lambda = 10$ and $H_\lambda = 30$. For the rollout stage, we use a temperature of 1.0 and a top-p value of 1.0. We set the KL loss coefficient to 0.001 and entropy loss coefficient to 0.001. All the models are trained with 8 H100 GPUs. We set the training mini-batch size as 256 and allow the models to generate 4096 tokens per prompt.

### C.2  Prompt Template

We present the template used for LLM to compare step-level reasoning.

---

**Prompt for Finding Reasoning Flaws in Correct Response via LLM-as-a-judge**

Here is the problem and the assistant's solution, which has been broken down into `{step}` steps.

**Problem:**

**Assistant's Solution:**

Your task is to review each step of the solution in sequence, analyzing, verifying, and critiquing the reasoning in detail. You need to provide the analyses and the conclusion in the following format:
<step>Step 1 Analysis</step>
<step>Step 2 Analysis</step>
... [CONTINUE FOR ALL step steps in the Assistant's Solution] ...
<conclusion>Correct/Incorrect</conclusion>

- When you analyze each step, you should use proper verification, recalculation, or reflection to indicate whether it is logically and mathematically valid. Please elaborate on the analysis process carefully.

- If an error is detected in any step, you should describe the nature and cause of the error in detail, and suggest how to correct the error or the correct approach. Once a step is found to contain any error, stop further analysis of subsequent steps (as they may depend on the identified error) and directly provide the conclusion of "Incorrect."

For instance, given a solution of five steps, if an error or flaw is found in the third step, you should reply in the following format:
<step>Step 1 Analysis</step>
<step>Step 2 Analysis</step>
<step>Step 3 Analysis; since an error or flaw is found here, also provide detailed critique and correction guideline)</step>
<conclusion>Incorrect</conclusion>
Respond with your analyses and conclusion directly.

---

> **Prompt for Responses Comparison via LLM-as-a-judge**
>
> **System** You are a meticulous, comparison engine. Your ONLY function is to compare the intermediate reasoning steps of the two responses provided to you.
> **User** Here is the problem and assistants' two solutions, which have been chunked into steps. You MUST provide preference over the two solutions.
> Problem: <prompt>
> Assistant's Solution 1: <solution1>
> Assistant's Solution 2: <solution2>
>
> Both solutions are correct. You MUST compare them based on the following criteria:
>
> - The reasoning process is more correct, and logical.
>
> - The reasoning process does not skip any reasoning steps.
>
> - The reasoning process does not skip any reasoning steps.
>
> You MUST follow this exact format:
> Your detailed verification reasoning goes here. Conclude with the number of the preferred solution: 1 or 2.
> If you prefer solution 1, you MUST output 1.
> If you prefer solution 2, you MUST output 2.
>
> Your preference:

## D ADDTIONAL EXPERIMENTAL RESULTS

In this section, we include additional ablation studies and evaluation results for a more comprehensive understanding of the PROF-GRPO framework.

### D.1 EFFECT OF ROLLOUT NUMBERS

We study the scale of rollout numbers $n$ with fixed policy-update number $m = 4$ by varying $n = 4, 8, 12, 16$. The lower-right plot in Figure 5 presents the test accuracy averaged over all five benchmarks for PROF-GRPO (Both) and Filter-Correct (Correct) started from Qwen2.5-Math-7B-base. We observe the performance first increases then decreases as $n$ increases, revealing a trade-off between enhancing process reasoning quality and avoiding reward hacking. Notably, Filter-Correct decreases later (after $n = 12$) because it only leverages the influence of PRM only in the correct group, indicating that Filter-Correct is more robust when the PRM's influence is higher, like when increasing the scale of ranking and filtering.

### D.2 VARIANTS OF FILTRATION METHODS

| Algorithm | Math500 | Minerva Math | Olympiad Bench | AIME24 | AMC23 | Average |
|-----------|---------|--------------|----------------|--------|-------|---------|
| Mean | **83.1** | **39.0** | **47.8** | 17.5 | **70.9** | **51.7** |
| Minimum | 82.9 | 38.3 | 46.7 | 20.8 | 65.9 | 50.9 |
| Sum | 82.4 | 38.1 | 47.4 | 17.7 | 67.5 | 50.6 |
| Ratio | 81.4 | 36.6 | 45.0 | **24.8** | 65.2 | 50.6 |

Table 4: Performance of different filtration ways in PROF starting from Qwen2.5-Math-7B-base.

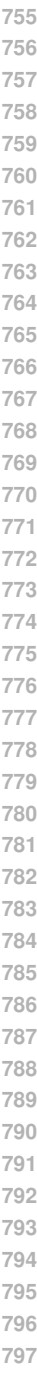

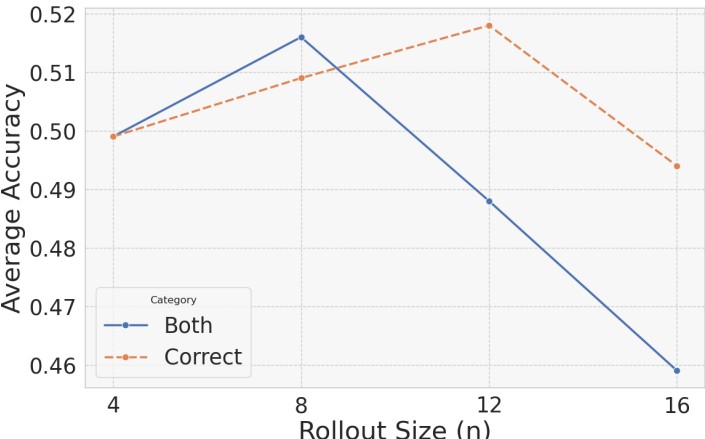

Figure 5: The averaged accuracy across all five benchmarks over rollout sizes $n = 4, 8, 12, 16$ for filtering both correct and incorrect groups with PRM consistency (Both) and only the correct group with PRM consistency (Correct).

In this subsection, we investigate the influence of different computation methods of consistency score $r^{\text{pro}}$ in addition to the mean of PRMs over steps, where Mean denotes averaging over steps in Algorithm 1, Minimum and Sum denotes taking the minimum and sum summation over steps, Ratio denotes filtering while preserving the original positive–negative sample distribution, instead of balancing. As shown in Table 4, the performances of Minimum (50.9%), Sum (50.6%), and Ratio (50.6%) are inferior to the mean. This suggests that the mean provides a more stable estimate of reasoning consistency: unlike the minimum, it is less sensitive to a single poorly scored step, and unlike the summation, it avoids bias towards longer trajectories. Additionally, balancing the correct-incorrect ratio can use data consistency to select the group with more sufficient samples without breaking their balance.

### D.3 EFFECT OF STEP NUMBER

To prove that PROF effect not by simply increasing the step number, We conduct the Filter-Nstep: Ranking and filtering out the samples with smaller number of steps instead of lower PRM-ORM consistency.

From Table 5, we find that Ratio scores 51.7% on average and cannot compete with balancing the proportion (PROF-GRPO), which also corroborates the conclusion that maintaining a balanced correct-incorrect proportion is essential. Additionally, since we observe that PROF boosts the number of intermediate reasoning steps, to verify that PROF does not simply increase the step length, but more importantly, enhances the quality of reasoning steps, we simply use the step length as the filtering criterion for comparison. As shown in Figure 6 and Table 5, Filter-Nstep manipulates the step length, which exhibits an unreasonable increase followed by a sudden drop, and its average accuracy is inferior.

## E ADDITIONAL EXAMPLES

| Algorithm | Math500 | Minerva Math | Olympiad Bench | AIME24 | AMC23 | Average |
|-----------|---------|--------------|----------------|--------|-------|---------|
| PROF-GRPO | **83.1** | **39.0** | **47.8** | 17.5 | **70.9** | **51.7** |
| Filter-Nstep | 81.5 | 35.5 | 45.9 | 16.3 | 58.6 | 47.6 |

Table 5: Performance of variants of filtering methods besides PROF-GRPO starting from Qwen2.5-Math-7B-base averaged on all five benchmarks, where Ratio denotes filtering while preserving the original correct/incorrect proportion, Filter-Nstep denotes ranking and filtering based on the number of step segments.

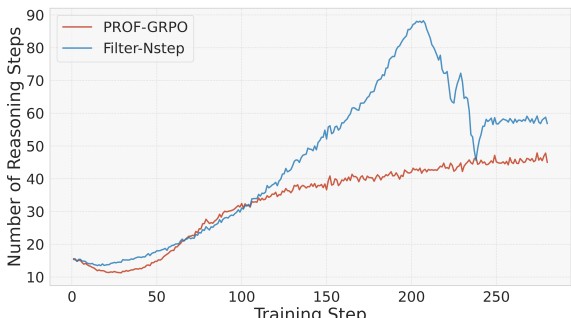

Figure 6: The number of reasoning steps during training time for PROF-GRPO and Filter-Nstep initialized from Qwen2.5-Math-7B-base.

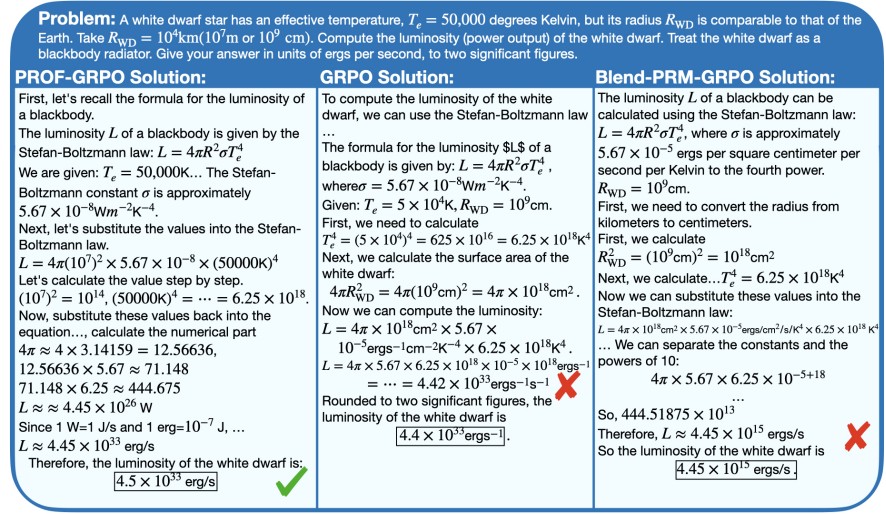

Figure 7: A Minerval-Math example to compare distinct intermediate reasoning patterns of PROF-GRPO, vanilla GRPO and Blend-PRM-GRPO. PROF-GRPO presents concrete and correct deduction steps. GRPO's solution skips detailed deduction steps and there are flaws in the calculation precision and the final rounding. Blend-PRM-GRPO has long-winded steps and makes a big mistake in calculating the power.

**Problem:** The proper divisors of 12 are 1, 2, 3, 4 and 6. A proper divisor of an integer $N$ is a positive divisor of $N$ that is less than $N$. What is the sum of the proper divisors of the sum of the proper divisors of 284?

**PROF-GRPO Solution:**

First, we need to find the proper divisors of 284 …
Let's start by finding the divisors of 284. We can start by checking the obvious small divisors.
Let's check if 2 is a divisor of 284.
$$284 \div 2 = 142$$
Since the result is an integer, 2 is a divisor.
Next, let's check if 4 is a divisor of 284…
The proper divisors of 284 are 1, 2, 4, 71, and 142.
Next, we need to find the sum of these proper divisors. Let's add these step by step …
$$1 + 2 + 4 + 71 + 142 = 220$$
Now, we need to find the proper divisors of 220. …
The proper divisors of 220 are 1, 2, 4, 5, 10, 11, 20, 22, 44, 55, and 110. The sum is:
$$1 + 2 + \ldots + 110 = 284$$
Therefore, the sum of the proper divisors of the sum of the proper divisors of 284 is $\boxed{284}$.

**GRPO Solution:**

First, we need to find the proper divisors of 284. The proper divisors of 284 are the positive divisors of 284 that are less than 284. The divisors of 284 are 1, 2, 4, 71, and 142. The sum of these proper divisors is:
$$1 + 2 + 4 + 71 + 142 = 220$$
Next, we need to find the proper divisors of 220. The divisors of 220 are 1, 2, 4, 5, 10, 11, 20, 22, 44, 55, and 110. The sum of these proper divisors is:
$$1 + 2 + \ldots + 110 = \boxed{284}$$
So the sum of the proper divisors of the sum of the proper divisors of 284 is 284 .

**Blend-PRM-GRPO Solution:**

First, we need to find the proper divisors of 284. Let's start by finding all the divisors of 284. We'll start with 1, which is always a divisor. We check if 2 is a divisor.
$$284 \div 2 = 142$$
So, 2 is a divisor.
Next, we check if 3 is a divisor.
$$284 \div 3 \approx 94.67$$
So, 3 is not a divisor …
We continue this process until we reach 142. The divisors of 284 are 1, 2, 4, 71, 142.
Now, we need to find the sum of these proper divisors: $1 + 2 + 4 + 71 + 142$
First, we add 1 and 2: $1 + 2 = 3$. ….
So, the sum of the proper divisors of 284 is 220.

Now, we need to find the proper divisors of 220. We'll check each number from 1 to 219 to see if it divides 220 without leaving a remainder …
The proper divisors of 220 are 1, 2, 4, 5, 10, 11, 20, 22, 44, 55, and 110….
So, the sum of the proper divisors of 220 is $\boxed{284}$.

Figure 8: A Math500 example to compare distinct intermediate reasoning patterns of PROF-GRPO, vanilla GRPO and Blend-PRM-GRPO. PROF-GRPO presents concrete and correct deduction steps. PROF-GRPO's solution shows how to find the divisors and summation in detail, and is easy to follow. GRPO skips all core reasoning. Blend-PRM-GRPO has inefficient and excessively tedious steps.