# OpenReview forum: "Beyond Correctness: Harmonizing Process and Outcome Rewards through RL Training"
_ICLR.cc/2026/Conference — ICLR 2026 Conference Withdrawn Submission_

### Official Review · Reviewer_8bmv · 2025-10-14

**Soundness:** 2
**Presentation:** 2
**Contribution:** 1
**Rating:** 2
**Confidence:** 4

**Summary:**

The paper proposes a function to combine process reward and outcome reward rather than adding them. The experimental results show that their combination algorithm outperform naive blending.

**Strengths:**

The empirical experiments show their filter algorithm can outperform GRPO and simply combining outcome rewards and process rewards.

**Weaknesses:**

> Limited contribution
> ---
> The only contribution of this paper is to heuristically design a new function that combines outcome reward and process reward. The function design lacks theoretical analyses and insightful motivation.

> Doubtful extensibility
> ---
> The experiments are only conducted with one PRM Qwen2.5-Math-PRM-7B. Whether their function can work with other PRMs is doubtful.

>Typos:
>---
>- L121:  blend PRMs and PRMs -> blend PRMs and ORMs
>- L198 (algorithm L4): K_+ -> k_+

**Questions:**

See weakness.

---

### Official Review · Reviewer_t6Pa · 2025-10-29

**Soundness:** 2
**Presentation:** 3
**Contribution:** 2
**Rating:** 2
**Confidence:** 4

**Summary:**

This paper addresses limitations in RLVR for mathematical reasoning, where ORMs provide coarse supervision and PRMs offer fine-grained but noisy feedback prone to hacking. The authors propose PROF, a data curation framework that oversamples rollouts, computes PRM-ORM consistency scores with step-length regularization, and filters by ranking correct and incorrect groups separately to balance training data and eliminate noisy gradients.

**Strengths:**

1. By using PRMs only for ranking/filtering rather than direct gradients, PROF avoids entropy collapse and verbose over-generation seen in blending methods, enabling stable, faster convergence.
2. Demonstrates gains in both final accuracy (e.g., 51.7% vs. 49.9% on Qwen-7B) with qualitative examples showing more verifiable, detailed cots.
3. Separates correct/incorrect groups for balanced variance maximization, with thorough ablations (e.g., rollout n=8 optimal) providing actionable insights for RLVR data curation.

**Weaknesses:**

1. Since PROF performs more aggressive filtering on rollout samples, it raises concerns on computational overhead. Oversampling (n=8 vs. 4) doubles rollout costs per iteration, which is unquantified against baselines, potentially limiting scalability for larger models or datasets.
2. The baseline description is vague. The authors cited 3 papers and summarize their methods as Blend. However, their methods are not the same. What exactly is the baseline and how the authors implement it remains unclear.
3. Performance hinges on pre-trained PRMs (e.g., Qwen2.5-Math-PRM-7B); Compared with GRPO, it may serve as an unfair advantage.
4. As shown in Figure 1, only 30% filtered correct samples are recognized as problematic, Does this means the judge accuracy is low? The authors should report the flaw ratio among the full set of the correct samples.
5. Lack of scaling experiments. Main experiments are only conducted on 3B/7B models with 7B PRM. I wonder if the advantage remains when using a large policy model with small PRM.
6. In ablation study, it seems PROF underperforms PROF-Correct and matches the original GRPO.

**Questions:**

- Did you use filtering for baseline GRPO? It is a common practice to filter out all-correct/wrong samples.
- In MC estimation, what do you mean by "from that point"? What is MC Estimation Accuray and how is it related to intermediate step value?

---

### Official Review · Reviewer_XiQh · 2025-11-01

**Soundness:** 2
**Presentation:** 3
**Contribution:** 2
**Rating:** 4
**Confidence:** 4

**Summary:**

This paper proposes PROF, a process consistency filtering approach that harmonizes ORMs with PRM through consistency-driven sample selection, achieving superior RL training performance compared to traditional ORM or PRM methods.

**Strengths:**

1. ROF demonstrates consistent improvements over both GRPO and naive blending approaches, with clear gains across benchmarks.
2. The analysis reveals that PROF effectively reshapes the model's Chain-of-Thought process from unfaithful reasoning into detailed and easy-to-verify steps, providing more insights into the how PRMs could be helpful.

**Weaknesses:**

1.  Despite experimental improvements, the method's complexity raises questions about scalability. Experiments are limited to ≤7B models (e.g., Qwen 7B, known for poor initialization, where performance gains are easily achieved). The PROF-GRPO improvements over GRPO appear relatively marginal given this context, (but the method requires an additional PRMs and more complicated training set up)
2. The PRM-ORM integration appears to be empirical reward shaping without a strong theoretical justification.  I worry hand-designed consistency mechanism may be task-specific rather than fundamentally superior.

**Questions:**

Can authors provide a theoretical analysis explaining why this particular PRM-ORM mixing approach is essential and irreplaceable, rather than merely empirically effective?  How does shaping PRMs and combining ORM affect the convergence point/ optimization gradient of the algorithm?

---

### Official Review · Reviewer_iV1q · 2025-11-02

**Soundness:** 2
**Presentation:** 3
**Contribution:** 2
**Rating:** 4
**Confidence:** 4

**Summary:**

This paper proposes a method to harmonize Outcome Reward Models (ORMs) and Process Reward Models (PRMs) for RL training in reasoning tasks. The goal is to address the limitations of ORMs (which are accurate but provide sparse supervision, potentially rewarding correct answers derived from flawed processes) and PRMs (which are dense but often noisy and prone to reward hacking).

Instead of directly blending the numerical rewards, the proposed method, PROF (Process consistency Filter), uses the consistency between ORM and PRM signals to filter training samples. Specifically, it retains correct (ORM-positive) samples that also have high PRM scores, and incorrect (ORM-negative) samples that also have low PRM scores. The experiments demonstrate that RL training on this filtered dataset can improve both final answer accuracy and the quality of intermediate reasoning steps.

**Strengths:**

- The paper addresses a core and timely challenge in LLM reasoning: the trade-off between the accuracy of sparse ORMs and the supervision density of noisy PRMs.

- The proposed method is simple and intuitive. Using consistency as a filter rather than directly blending reward values is a clever approach to leverage the strengths of both signals while mitigating their weaknesses.

- The ablation studies effectively demonstrate the necessity of separating the correct and incorrect groups for filtering. Furthermore, the evaluation of intermediate step quality (using MC estimation and LLM-as-a-judge) provides good evidence that the method's benefits extend beyond just final-answer accuracy.

**Weaknesses:**

- The method's primary weakness is its apparent reliance on a high-quality PRM. The paper does not provide a quantitative analysis of the PRM's capabilities or how the PROF method's performance scales with the quality of the PRM. This makes it difficult to assess the method's true requirements and generalizability.

- The main experimental results are heavily skewed towards the Qwen model family. Table 2, which presents the main results, is entirely missing results for other widely-used model families like Llama. Given the known diversity in model behaviors, validation on Qwen models alone is insufficient to make broad claims, as has been shown in other recent work. The generalizability of this method is therefore not well-established.

**Questions:**

- How sensitive is the PROF method to the quality of the PRM? What would the results look like if a significantly weaker (or stronger) PRM were used for filtering? Is there a "sweet spot" for PRM quality where this method is most effective?

- The results in Table 3 (LLaMA-3.2-3B) show that "PROF-GRPO (Correct)" (25.4%) outperforms "PROF-GRPO (Both)" (23.9%). Does this not contradict the main claim that filtering both groups is optimal? This finding suggests that the conclusion regarding the necessity of filtering the incorrect group may not be robust, and might even be detrimental when generalizing to other model families. Can the authors elaborate on this?

---

### Note · Authors · 2025-11-20

I have read and agree with the venue's withdrawal policy on behalf of myself and my co-authors.